# Computational Modeling of the Human Microbiome

**DOI:** 10.3390/microorganisms8020197

**Published:** 2020-01-31

**Authors:** Shomeek Chowdhury, Stephen S. Fong

**Affiliations:** 1Integrative Life Sciences, Virginia Commonwealth University, 1000 West Cary Street, Richmond, VA 23284 USA; schowdhury2@mymail.vcu.edu; 2Chemical and Life Science Engineering, Virginia Commonwealth University, 601 West Main Street, Richmond, VA 23284, USA

**Keywords:** human microbiome, genome scale modeling, DNA sequencing, human health

## Abstract

The impact of microorganisms on human health has long been acknowledged and studied, but recent advances in research methodologies have enabled a new systems-level perspective on the collections of microorganisms associated with humans, the human microbiome. Large-scale collaborative efforts such as the NIH Human Microbiome Project have sought to kick-start research on the human microbiome by providing foundational information on microbial composition based upon specific sites across the human body. Here, we focus on the four main anatomical sites of the human microbiome: gut, oral, skin, and vaginal, and provide information on site-specific background, experimental data, and computational modeling. Each of the site-specific microbiomes has unique organisms and phenomena associated with them; there are also high-level commonalities. By providing an overview of different human microbiome sites, we hope to provide a perspective where detailed, site-specific research is needed to understand causal phenomena that impact human health, but there is equally a need for more generalized methodology improvements that would benefit all human microbiome research.

## 1. Introduction

Interest in microorganisms associated with the human body has a long history dating back to the handcrafted microscopes built by Antonie van Leewenhoek in the 1670s, where bacteria in van Leeuwenhoek’s oral and fecal samples were referred to as “animalcules” [1]. As research techniques and knowledge increased, human-associated microorganisms continued to be studied (e.g., use of Hungate tubes to isolate anaerobic microbes [2] and documented (e.g., book titled “A Flora and Fauna within Living Animals” in 1853). The accumulation of research has led to the current concept of the human microbiome that has been described as an ecological community of commensal, symbiotic and pathogenic microorganisms that literally share our body space and have been all but ignored as determinants of health and disease [3].

Early glimpses into the complexity of the human microbiome started in the 1980s where diverse microorganisms such as *S. aureus*, *E. coli* and *Viridans* were found by Rotimi et al. in the umbilical, oral and fecal flora of neonates [4]. More recently, Pasolli and co-workers examined the microbiomes at the four human anatomical sites: stool, vagina, skin and oral, using large-scale metagenomic approaches and found ~150,000 microbial genomes coming from 4930 species (~total number of species in the human microbiome) [5]. Interestingly, 77% (3796) of the identified species were novel species whose genomes were not present in any of the public repositories. The scope and novelty of composition was further emphasized where it was noted that ~75% genes associated with the human microbiome lack functional annotation meaning that there is a large amount of “functional dark matter” associated with the human microbiome [6]. Apart from genes, proteins and metabolites in the human microbiome, recently, researchers have found that human microbiome is not restricted to only microbes and their metabolites, genes and enzymes. Small proteins or microproteins (less than 50AA) have been regularly found in human microbiome with the possibility of millions of small open reading frames (ORFs) or microproteins being present [7]. The possible functions of the microproteins include: housekeeping, bacterial adaptation, bacterial defense against phages and microbe–microbe and microbe–host cell communication. These types of novel findings indicate the mysterious side of the human microbiome. Ursell et al. and Gilbert et al. have already suggested that human microbiome is still highly unexplored [8,9].

16S rRNA sequencing has been the standard and regular approach to find the species composition of the human microbiome [10]. The hypervariable regions V1–V3 and V3–V5 of the 16S rRNA gene help in identifying the taxonomic composition of various bacterial species. People have clustered this gene into operational taxonomic units (OTUs) to investigate the microbiota composition in healthy humans [11]. Sanger sequencing has been the standard method to sequence the complete stretch of the amplicon (16S rDNA) [12]. However, people realized that species composition can be identified using shorter DNA stretches with higher sequence coverage and thus next generation sequencing (NGS) technologies, i.e., Roche 454 pyrosequencing, Illumina and Ion Torrent sequencing [12] are also used for meta-genomic sequencing. Later, many computational approaches have also been developed to analyze the 16S rRNA sequences of both disease and non-disease causing microbes to better understand their biology in the human microbial communities [10,13]. However, even if we have good coverage and longer sequencing reads using 16S rRNA sequencing, it will always be hard to obtain the genomic information of low abundance species [6,10]. Therefore, recent research has shifted to using high-throughput data techniques to produce both the qualitative and quantitative knowledge of the DNA, mRNA transcripts, metabolites, and proteins of the microbial groups in the microbiome [14,15]. Meta-omic approaches can help provide a more comprehensive functional view of microorganisms and their roles within the microbiome. Shotgun metagenomic sequencing was the first step in this direction where bacteria’s whole genomic DNA from human/environmental samples is analyzed for both species identification and understanding gene function potential of the microbe [12,16]. Another example is the HMP Unified Metabolic Analysis Network (HUMAnN) that performs metabolic and functional reconstructions of metagenomic data [17]. This method was applied on seven primary human body sites including stool, tongue dorsum and anterior nares on 102 individuals. They identified the core metabolic pathways, genes and functional modules which were different for different sites across individuals. In the vaginal microbiome, it was found that glycosaminoglycan degradation, phosphate and amino acid transport are more active in this microbiome [18].

Building upon the increased experimental data generated through the high-throughput approaches, computational modeling approaches such as genome scale metabolic models (GEMs) have been developed to integrate and analyze data to study function (Figure 1). In recent years, meta-omics data have been used in conjunction with genome scale metabolic models (GEMs). This is illustrated in Table 1, where omics and meta-omics data were used in the majority of the GEM studies. Genome scale metabolic models and metagenomics data is taken as an input when you are using MAMBO (Metabolomic Analysis of Metagenomes using fBa and Optimization) [19]. This research study used this approach where they incorporated 1500 microbes in their model and showed that a distinct metabolome exists at vagina, stool, skin and oral sites in the human body [19]. Use of in vitro, ex vivo and in vivo experimental data with in silico models serve as the excellent research pipeline to discover the unknown microbe–microbe and microbe-host metabolic interactions in human microbiomes suggesting crucial therapeutic advancements [14]. While each of the respective omic data types provide useful information in characterizing organism function, some of the data types are more directly converted to the modeling formalism than others. For example, Vanee et al. used a proteomics derived model to understand the metabolism functionalities of the microbe *Thermobifida fusca* where the growth rates shown by experimental and in silico data were almost similar [20]. If experimental and computational work is properly used collaboratively, there will be identification of not only the representative species of the human microbiome but of the other unknown species with whom these leader microbes coordinate through vast number of metabolite exchanges [21].

Computational analyses such as network analysis, agent-based modeling, and genome scale metabolic modeling (GEM) have used to study various aspects of the human microbiome including structure, dynamics, and coordinated function [22,23,24]. While a variety of specific analyses are encompassed in network analysis, generally this approach considers connectivity of components to consider structure-function relationships. This type of analysis often leads to the identification of critical, highly connected hubs/nodes and can provide insight into robustness to perturbations. Agent-based modeling is a stochastic simulation approach where discrete agents are ascribed attributes to represent a biological entity and allowed to interact dynamically with other agents. For example, agents could be used to depict an enzyme and a substrate and different concentrations of these could be simulated to study the time-course dynamics by changing binding affinities. Thus, system function can be studied by modifying properties of an agent and running simulations to see effects. Genome scale metabolic modeling is a modeling approach that formulates a model based upon experimentally-established information such as gene content (genomic data) and biochemistry, and a number of publications and reviews exist describing this method in detail [25]. By starting with gene content and connecting associated enzymes with biochemical function, a stoichiometric matrix is generated that allows simulations to be run that can be used to study a variety of biological systems including human–microbe and microbe–microbe interactions [26,27]. A number of software packages have been developed to support development and analysis of genome scale metabolic models including the constraint based reconstruction and analysis (COBRA) Toolbox [28]. Reconstructed networks for an organism represent the biochemical and genetic capabilities and can be analyzed by stipulating input constraints to create a space of allowable flux distributions to elucidate all of the possible metabolic flux states. Flux balance analysis (FBA) is the generic approach for calculating the flux/flow of metabolites through the network in an organism and hence, predicting the growth phenotype of the organism or rate of production of an important chemical compound [29]. It finds the optimal solutions to the objective function which basically depicts the biological function which the network is performing. For example, if you want to predict growth, biomass production by the target organism is taken as the objective function [30]. After predictions are made using FBA, validation with experimental data or model reconciliation is achieved between experimental and computational models using various algorithms [31,32,33]. This entire process of building, analyzing, and testing GEMs helps obtain a broad functional understanding of an organism and subsequently helps analyze and predict difference between organisms, for example, between pathogens and non-pathogens of gut microbiome to find therapeutic targets [34].

Given that the human microbiome involves numerous interacting species, community genome scale metabolic models are definitely required for capturing the microbiome biology in a comprehensive manner [26,28,35]. However, community genome scale modeling has challenges that need to be addressed. First, most of these studies have been done on individual species because the number of species of the human microbiome which have models is very low (~25) [21,28,35]. In the biological world, there are of course not one or two, but multiple species working either in a cooperative or competitive way in the complex microbiome. This brings us to the second challenge which is specifying a global objective function for the whole community of microbes. People, however, have constantly made efforts to tackle this problem by designing constraint based/genome scale modeling software packages. A python software tool called Micom was developed which can take into consideration the objective functions for both individual species and the microbial community including ~100 species at a time [36]. Baldini et al. have created a MATLAB based software called Microbiome Modeling Toolbox which applies genome scale modeling to simulate pairwise microbe–microbe and human–microbe interactions [27]. Compartmentalization is the third challenge where reactions and the metabolites in microbiome are needed to be partitioned correctly in the appropriate compartments (species or cell organelles) [6,35]. 

There is microbial diversity across each of the human microbiome sites (gut, oral, skin and vagina) and temporal diversity meaning varied biogeography dynamics exist for each of these sites which advocates a more directed research towards personalized medicine for combating the human microbial diseases [8,9,37]. The microbial behaviors and the molecular mechanisms at these different human anatomical areas is different from each other [5,8,9]. The vast and complicated nature of the human microbiome (the microbiome between two individuals is 80%–90% different compared to the human genome which is 99.9% similar among individuals [8]) has even given rise to a new multidisciplinary field of systems microbial medicine where experts from a wide range of fields like microbiology, genetics, mathematics, statistics, engineering, computational biology, nutrition, immunology, neurology and endocrinology are required to work together to get insights to human microbiome [38]. The Human Microbiome Project (HMP) was a great initiative in this direction where experts from different areas are trying to investigate different aspects of human microbiome like species composition, metabolome, microbe–microbe interactions [39]. 

Studying the human microbiome is a challenging endeavor, but the connection to human health is becoming clearer as the four major human microbiomes (skin, oral, gut, and vaginal) have their own associated diseases: Crohn’s disease and obesity in gut, bacterial vaginosis and preterm birth in vaginal, periodontitis in oral, atopic dermatitis in skin [40,41,42,43,44]. Therefore, there is a huge need to study the human microbiome at these four anatomical sites in depth. Our review covers the background, high throughput studies, and modeling methodologies employed to study each of the four distinct microbiomes (gut, oral, skin and vaginal). Bringing all of them under one banner will help us to get a holistic view of the global human microbial interactions which can be used in future for the development of effective and novel treatment strategies.

## 2. Human Gut Microbiome

### 2.1. Background

The longest established human microbiome had foundations in early work identifying microorganisms associated with the gastrointestinal tract. Current concepts of the gut microbiome dates back to 1965 when people found that gut microbes are not only present in the large intestine but also in the stomach and small intestine [45]. Early research in this area mainly utilized culturing techniques where researchers isolated the bacteria from the GI tract (*Lactobacilli*, *Streptococci, Bacteroides*) and could test effects by feeding germ-free mice with a diet contaminated with isolated bacteria [46]. Zoetendal et al. in 1998 using temperature gradient gel electrophoresis of 16S rRNA proved that every human has their own distinctive fecal microbiome [47]. Yatsunenko et al. showed that gut microbiome is unique for every individual by characterizing the bacterial species from fecal samples of individuals of different ages, living in different locations and also having different cultural traditions, diet and lifestyle [48]. External factors including diet, antibiotics, lifestyle and even occupation alter microbial niches to a great extent. Apart from that, the dynamics of a diseased and healthy microbiome are totally different. Obesity and diabetic patients have often witnessed a different gut microbiome from healthy individuals where diet has been seen to change the microbiome dynamics hugely [49]. Research has also been conducted to understand how geography influences the gut microbiome. For example, a comparison of the microbiomes of Nigeria and UK infants, small children and adults was conducted [50]. Drastic differences in the species composition between these two countries were identified and diet noted as a factor affecting the microbiome composition, however, diet had a minimal effect in altering the microbiome of adults [50]. It was suggested that the microbiome which gets established during childhood by the diet taken at that age will not change hugely as the person grows. A major transition in microbiome composition occurred in the growth of infants to small children that corresponds to the switch from liquid to solid food, thus there are multiple contextual and temporal variables that can affect the microbiome [50].

Compositionally, there are often characteristic, dominant organisms associated with the gut microbiome; however, low abundance species may play crucial roles as well. Low abundance microbes follow a lottery-based schema where a single niche is occupied by a single species and other species are completely excluded [51]. Dominant phyla like *Bacteroidetes* and *Firmicutes* do not exhibit competitive lottery-based schema whereas low abundant species (*Akkermansia, Dialister*, and *Phascolarctobacterium* genera) do. There is a restricted functioning of the metabolome of low abundance microbes as they typically have a smaller number of total genes and do not have clear patterns of coexistence or resource partitioning. Another study focused on low abundance microbes where they injected 11 strains of commensal bacteria into mice [52]. These rare strains protected the mouse from foreign microbes like *Listeria monocytogenes* and they were also responsible for producing anti-tumor immunity by causing development, maturation, accumulation and induction of IFN_gamma+ CD8 T cells in the gut. Overall, it has generally been found that a high diversity gut microbiome is favorable over a low diversity gut microbiome.

Through various studies, it has been found that the gut microbiome is responsible for performing metabolic functions which regulate host functions such as bile acid metabolism where the host synthesizes bile acids and the deconjugation and transformation of bile acids is performed by the gut microbes [40]. It was further found that the gut microbiome gets depleted in people with Inflammatory Bowel Disease (IBD) compared to healthy patients, affecting bile acid metabolism.

Additional health effects of the gut microbiome are related to interactions with the immune system. The gut microbiome has been shown to perform defense mechanisms against the foreign pathogens called “colonization resistance” [53] where the microbiota protects the host from infection. The probiotic strain *Lactobacillus salivarius* UCC118 protects the host from *Listeria monocytogenes* [53,54]. Given the protective role of the gut microbiome, pathogens have developed strategies to target the microbiome as an infection mechanism. *Citrobacter rodentium* and *Salmonella* modify the host’s gut microbiome for their invasion by causing intestinal inflammation [54,55,56]. The knowledge of the defense mechanisms by the foreign pathogens against host’s colonization resistance will definitely have significant medical importance. Antibiotics also alter the gut microbiome significantly affecting the host physiology to a large extent [57,58]. Consuming antibiotics when not required has been related to several diseases like asthma and type 1 diabetes [59,60]. Other studies have also found new immune mechanism which human microbiome show. The microbial metabolite short-chain fatty acids produced by commensal bacteria have been shown responsible for differentiation and generation of regulatory T cells whose main function is to maintain immune homeostasis [61,62]. Salicyl-azo-sulfapyridine (SAS) is a drug which the intestinal microbiome metabolizes by reducing the azo bond [63]. This activity was significantly reduced when neomycin was injected into rats which inhibits the intestinal microbiome functioning. It has been also found that gut microbiome provides an important path between cancer cells and immunotherapy treatment [64]. Commensal gut microbes cause inflammation, produce more TNF from the myeloid cells and modulate the increased production of radical oxidative species (ROS) from myeloid cells which results in effective immunotherapy treatment.

In the context of cancer, a highly diverse gut microbiome with dominant phyla of *Ruminococcaceae* and *Faecalibacterium* showed effective anti-tumor response and improved the T-cell function in the tumor microenvironment to anti-PD1 immunotherapy in melanoma patients [65]. A low diversity gut microbiome with dominant phyla *Bacteroidetes* did not show any protective mechanisms against cancer.

Shift in the dominant phyla composition in the gut microbiome is an aspect which is very much associated with obesity [66]. The species composition, metabolome, microbial environment all gets altered during obesity. Another interesting property people have witnessed in gut microbiomes associated with obesity is energy metabolism [67] where microbiota associated with obesity have the ability to harvest more energy from the diet than in non-obese individuals. There is a need to understand the complex interplay between host genetics and microbial community gene content to investigate obesity. The biological properties of the ‘superorganism’ which forms when microbes and their genes, enzymes and metabolites and host’s genes, enzymes and metabolites interact together should be explored to get the insights to the obesity biology. 

### 2.2. High-Throughput Studies

High-throughput approaches, i.e., 16S rRNA sequencing and metagenomics have been used in many studies to identify the species composition in the gut microbiome. One of the ongoing challenges in microbiome work has been the issue of identifying low abundance species [68]. For example, using 16S rRNA sequencing, they found two gut microbes which are in low abundance in the human colon: *Lachnospira pectinoschiza* and another one similar to *Planctomycetes* (organism name was not known). Wilson and coworkers using 16S rRNA sequencing identified new species colonizing human colon [68]. Similarly, human intestinal flora diversity have been explored using the same method leading to the identification of many novel species where approximately 400 to 500 different species inhabit [69,70]. Furthermore, 16S rRNA sequencing studies found new metabolic pathways for glycans, vitamins and amino-acids in the gut microbiome [71]. 16S rRNA sequencing has also been used to determine the abundance levels of species in the gut microbiome during obesity. Ley et al. took samples of 12 obese people and observed that relative abundance of *Bacteroidetes* was high and of *Fimicutes* was low [72]. These 12 obese people were then given restricted carbohydrate and fat diets. After weight loss, the gut microbiomes of the 12 individuals were examined again using 16S rRNA sequencing and it was found out that the relative abundance of *Bacteroidetes* decreased and that of *Firmicutes* increased. Biological dynamics of the normal gut microbiome gets perturbed and altered during diseases like obesity, type-1 and type-2 diabetes and even atherosclerosis [41,73]. However, the limit in the read length in these kinds of studies sometimes misses the detection of some bacterial species and strains [8]. Although, this has been a long standing problem for now, people have already started making efforts to address this issue by utilizing recently developed sequencing technologies [74,75]. These are one molecule real time technologies which have been shown to decrease the short read problems and furthermore, a higher data yield at lower cost can be obtained by using this approach [76].

Recently, metagenomic studies have been employed to study microbiomes instead of 16S rRNA sequencing due to the amount of bacterial DNA it can analyze to identify new species. For example, a large study analyzed the gut microbiome samples of 11,850 people who live in South American and African regions [77]. In total, 92,143 metagenome assembled genomes were used and 1952 new bacterial species were found. Half of these bacterial species could not be classified even at the genus level, demonstrating the amount of unannotated information that exists. Similarly, Nayfach et al. analyzed 156,478 microbial genomes from fecal samples and found 23,790 species level operational taxonomic units (OTUs) [78]. Among that, 4588 OTUs were from the gut microbiome out of which 2058 OTUs were newly identified.

One of the challenges with metagenomic studies is associated with how to deal with novel species or genes that lack functional annotation. Binning is the principal approach in metagenomics for grouping the contigs to specific operational taxonomic units (OTUs) [79]. Zhu et al. used binning methods to identify the pathogenic species in the gut microbiome causing diarrhea [80]. Miller et al. suggested that conventional metagenomic protocols including assembly and binning which are performed using chemical and physical treatment of multiple tissue samples are unable to find low abundant species [81]. However, they modified and improved the existing metagenomic methods by focusing on single tissue sample without any external treatment and when applied to studying the invertebrate *Bugula neritina,* seven novel bacterial genomes were identified [81]. Another study has also tried to tackle these limitations of metagenomics by using targeted functional profiling as this experimental approach is known to identify bacterial species and their genes [82]. It was conducted in two manners: immunology based and non-immunology based. In the immunology based approach, bacterial response to IgA was evaluated to identify the species and the genes. In the non-immunology based approach, metagenomic DNA was expressed in a bacterial host and it was seen which genes are expressed and define the functional phenotype. People have predicted the function of the nonribosomal peptide synthetase (NRPS) gene that is serine protease inhibition using this method which they found in the cells of a number of gut microbes including *E. coli* and *B. subtilis.*


### 2.3. Modeling Methodologies

A number of different computational modeling approaches have been developed to study the gut microbiome to help analyze and understand its function including aspects of network topology, metabolic function, microbe-host interactions, and microbe–microbe interactions. One of the early computational studies of the gut microbiome used a network based approach where topological properties of metabolic networks were examined to analyze obesity and IBD [24]. It was found that the microbiome and associated metabolome has a large impact on host metabolism. Critical host enzymes of the metabolic network were found not on the central part of the network but on the periphery. It was suggested that these peripheral enzymes might be using bacterial metabolites which are unique and the compounds produced by these enzymes will get released to the host environment due to close proximity. Furthermore, these enzymes might be acting on the host metabolites to regulate the host environment.

Genome scale metabolic models (GEMs) have also been utilized to study various aspects of the gut microbiome including: microbe–microbe, host-microbe and diet-microbe metabolic interactions [83]. In addition to studying different interactions, GEMs have also been developed to analyze single species, multiple species and communities in the gut microbiome [84].

*E. coli* which is the common microbe found in gut microbiome, the strain specific differences of it have been examined using GEMs where metabolic reconstructions of the three common strains of *E. coli* (HS, UTI89, and CFT073) found in gut were compared [85]. The optimal flux distribution for each of the 3 strains were different but the optimal growth rate was the same and each strain had a distinct set of metabolic pathways explaining the reason for different strain of the same microbial species giving rise to a different phenotypic function. 

Additional application of single species GEMs associated with the gut microbiome consists of three main areas: (1) construction of in silico growth media; (2) phenotype prediction of the gut microbes; and (3) defining interspecies metabolic interactions [22]. In designing media, GEMs play a crucial role by identifying the metabolites required and how growth is computed. Transcriptomic and proteomics data is used by GEM to predict the phenotypes and explain the pathogenicity and therapeutics of the microbes even at the strain level.

Single species GEMs also serve as the foundation for studying community population function. One modeling approach is called mixed-bag network modeling [86]. This method takes the single species GEM and merges it into one model with a single intracellular compartment. People have used this methodology to understand the differences in the gut microbiome dynamics between malnourished children from Bangladesh and Malawi and healthy children from Sweden [87]. Gut microbiome is responsible for converting dietary macronutrients to health-promoting metabolites like vitamins, amino acids (AA) and SCFAs. GEM was performed for 58 commonly found gut microbes in both these populations and it was observed that gut microbiome dysbiosis is associated more with malnourished children. Essential AA like lysine, tryptophan, and arginine are produced in low amounts in malnourished children which act as precursors for many crucial metabolic pathways. GEMs showed that Bangladeshi and Malawi malnourished children had less species diversity resulting in less metabolome diversity and less essential AA production in comparison to Swedish healthy children.

Larger, multi-species genome scale metabolic modeling efforts such as CASINO and AGORA are also now available as well as community based approaches that utilized metagenomic data to study species co-occurrence as related to nutrient availability [88,89]. CASINO (Community And Systems-level INteractive Optimization) is a toolbox focused on community metabolic modeling [90]. In developing CASINO, five dominant gut microbes were used to study microbe-diet interactions. Assembly of gut organisms through reconstruction and analysis (AGORA) is the largest collection of gut microbiome GEMs accounting for 773 gut microbes coming from 605 different bacterial species [91]. AGORA produced metabolically distinct reconstructions for each organism and experimentally validated the carbon source uptake, fermentation product secretion and nutrient requirements of the gut microbes. Pairwise interactions in both aerobic and anaerobic conditions for all organisms were conducted. By also using the Recon 2 human metabolic model, it was demonstrated that human metabolism and gut microbiome metabolism were interlinked [92,93,94].

After construction of a GEM, omics data can be integrated into the model to analyze various phenomena associated with food-microbiota-gut dynamics [95,96,97]. The gut microbiome is the main link between body’s diet and nutrient uptake and it performs important metabolic functions like degradation of undigested polysaccharides and producing short chain fatty acids (SCFAs) [98]. Acetate, butyrate and propionate have been the major SCFAs which microbes produce and transport. GEMs have been used to study where SCFA transfer, ATP production and other growth strategies of the gut microbes. Shoaie et al. have used genome scale modeling (GEM) to model the metabolic interactions between the three dominant microorganisms of the gut microbiome; *Bacteroides thetaiotaomicron*, *Eubacterium rectale* and *Methanobrevibacter smithii* [99]. They integrated transcriptomic data with GEMs to evaluate the adaptation strategies of these organisms. SCFA production and consumption among the three microbes was studied the diffusion coefficient of SCFA absorption by epithelial cells was also calculated. Gut microbes mainly interact with the host through the exchange of bile acids, phenolic and aromatic acids, cholines, fatty acids and phospholipids and interact among themselves through SCFAs exchanges. Diet is the crucial factor in defining the function of the gut microbiome and metabolism of three main macronutrients: carbohydrates, proteins and fats by the gut microbiome can be studied using GEMs [100].

Another study investigated the effects of gut microbiome on host metabolism where GEM was used to study the co-metabolism of host and microbe under four in silico dietary conditions: High-protein, High-carbohydrate, High-Fat and Western [101]. 11 common gut microorganism models were used in conjunction with Recon 2 human metabolic model (human metabolism) [92]. It was shown that the gut microbiome produces essential metabolites leading to hormonal homeostasis. In another study, GEM was used to analyze the metabolic networks of infants before (1 and 3 months) and after (7 months and 1 year) solid food introduction [102]. The introduction of solid food precipitated changes in the gut microbiome where metabolites like ferulate were produced (responsible for cognitive development and neuroprotection).

To further facilitate research connecting the microbiome to the human host, the virtual metabolic human database (VMH) was developed [103]. The database used five resources: Human metabolism (Recon3D human metabolic reconstructions) [104], ReconMaps (seven human metabolic maps of six organelles and one accounting for all of them including the reactions occurring in extracellular space and cytosol) [94], Gut microbiome (818 manually curated GENREs of 667 species), Disease (255 mendelian diseases) and Nutrition (food database and diet database). They are linked through shared nomenclature of metabolites, reactions and genes. The idea is a metabolic disease is not caused by an individual factor but both external (diet, gut microbes) and internal (host genetics) factors are important to consider. The VMH database contains 17,730 and 5180 unique metabolic reactions and metabolites respectively, 632685 microbial genes and 3695 host genes, 255 diseases, 818 microbes and 8790 food items.

In addition to dynamics associated with food and species interactions, spatial and temporal dynamics are important to consider. Species in the gut sometimes support each other’s growth by metabolic cross-feeding while compete for the nutrients leading to the division of metabolic tasks. BacArena, a community modeling tool was developed and applied on seven commonly found species on the gut [105]. It integrates FBA and genome scale modeling (GEM) to look at microbial metabolism. Individual models of the microbes are considered where the spatial and temporal status of the metabolites in the gut microbiome are looked and then combined to understand the whole microbial community. An agent based model called GutLogo has also been developed to study the spatial and temporal dynamics [23]. Agent based model analysis showed that perturbation of the gut microbiome using nutrition and probiotics alters it but when the perturbations are removed, it again comes back to its original composition indicating the robust nature of this biological system. 

Modeling and analysis of the gut microbiome has helped gain insight into a variety of diseases as our understanding of the impact of the gut microbiome continues to expand. GEMs of 5 abundant gut microbes and the human small intestine and human brain were combined to study autism dynamics [106]. The model was combined with brain using PBPK transport model. The study was able to capture some biomarkers depicting autism like pyruvate and lactate. Two studies utilized GEMs of gut microbes to gain insights into the progression of colorectal cancer. The first study evaluated hydrogen sulfide production which was higher for tumor cells than normal cells during colorectal cancer [107]. 16S rRNA data was used as input to generate GEMs of the gut microbes *Fusobacterium nucleatum*, *Clostridium perfringens*, *Filifactor alocis* and *B. fragilis* which showed higher hydrogen sulfide production. GEMs showed higher flux for hydrogen sulfide and the metabolic interactions among these microbes was analyzed to understand the mechanisms of its production. A second study worked in a similar manner but the biological mechanism they focused on was the mismatch repair system (MMR) during colorectal cancer [108]. They created GEMs of gut microbes to understand the metabolic interactions in two types of microbial communities: dMMR (deficient) and pMMR (proficient). *B. fragilis* and *F. nucleatum* had higher abundances in dMMR colorectal cancer whereas *B. fragilis* had low abundance in pMMR colorectal cancer. GEM showed that hydrogen sulfide production was higher in dMMR colorectal cancer and there were amino acid substitutions in place of it in pMMR colorectal cancer produced in higher amounts. Different modeling studies performed on gut microbiome along with the experimental data used in that study during disease and non-disease scenarios is given in Table 1. The table also lists the modeling studies performed on other three microbiomes: oral, skin and vaginal microbiome.

## 3. Human Oral Microbiome

### 3.1. Background

In 1684 Leeuwenhoek wrote a letter to the Royal Society of London stating “The number of animals in the scurf of a man’s teeth is so many that I believe they exceed the number of men in a kingdom” [119]. As many as 700 microbial species reside in the human oral microbiome. The oral cavity is complex because of the different microbial niches allowing for diverse local environments. The different oral sites are teeth, the gingival sulcus, the tongue, mouth floor, cheek, hard and soft palates, tonsils, throat and saliva [119]. People have used different experimental approaches to investigate oral microbiome: culturing, microscopy, gel-based techniques, next generation sequencing (NGS), DNA microarrays, DNA hybridization, 16S rRNA sequencing and PCR [119,120]. The challenges that people face when they are dealing with oral microbiomes is bacteria in this microbiome form biofilms which is a complex microbial community [119,120]. They form stable metabolic interactions in biofilms which help them in their survival and growth. The biofilm formation helps the oral microbes to maintain homeostasis, prevent disease development and protect the oral cavity [120]. Apart from that, researchers have tried to find the reason behind the spatial organization of oral microbes i.e., if an oral microbe is present in a particular location in the oral cavity, why it is actually present there [121]. In a dental plaque microbiome, *Corynebacterium* is present in the biofilm matrix attached to the tooth as it holds the entire microbiome structure. *Streptococcus* is present in the periphery where they get an oxygen rich environment with access to sugars.

Biofilm disruption and oral microbiome imbalance causes diseases and dysbiosis in the oral microbiome [120]. Two widely known oral diseases; dental caries and periodontitis are caused by *Streptococcus mutans* and *Porphyromonas gingivalis* respectively. People have used NGS to identify novel species in both these diseases like *Granulicatella*, *Leptotrichia*, *Neisseria bacilliformis* in caries and *Prevotella spp.* in periodontitis [119]. It is not a single oral pathogen or group of pathogens which cause oral diseases but rather the collection of microorganisms. Apart from caries, periodontal and other mucosal diseases which are local oral diseases, imbalance in oral microbial flora has been linked to several whole-body systemic diseases like atherosclerosis, HIV, polycystic ovarian syndrome (PCOS), Alzheimer’s disease, inflammatory bowel disease (IBD), liver cirrhosis, diabetes and obesity [122]. Therefore, exploring these microbiomes will be helpful to develop human health improvement strategies as it shows effects on the whole body at various anatomical sites.

As with the gut microbiome, there is a large number of novel, unknown species in oral microbiome [120]. Dewhirst et al. created the Human Oral Microbiome Database (HOMD) where they performed phylogenetic analysis for the oral microbes at different locations from teeth to tonsils [123]. 16S rRNA gene sequence analysis identified 1179 taxa of which just 24% were named and 68% were uncultivated from 36,043 16s rRNA gene clones. To further support the scope of the oral microbiome, Benn et al. stated ~700 bacterial species are found in the oral microbiome out of which 50% of them have not been cultivated yet [119]. Of the known species and function, it has been found peptidic small molecules (PSMs) are common in the oral microbiome [124]. Oral microbes secrete and release PSMs and exchange it among themselves with other metabolites which are responsible for giving structure to the microbiome [124]. Even though PSMs can be detected using liquid chromatography-tandem mass spectrometry (LC-MS/MS), researchers noted that there is a huge “black box” in this area and deeper understanding of the oral microbial interactions is required to identify more PSMs and the processes by which they are differentially produced. These studies show that the oral microbiome is actually highly unexplored with a large number of unknown species and unique molecules, metabolites, metabolic genes and pathways. 

### 3.2. High-Throughput Studies

16S rRNA sequencing has been the most popular high-throughput approach to identify novel members in the oral microbiome [125]. *Rothia* are generally found at tongue or tooth surfaces, *Streptococcus salivarius* colonizes the tongue, *Simonsiella* colonizes mainly the hard palate and *Treponemes* can be found only at the subgingival crevice. The oral microbiome diversity is huge as 400–500 different bacterial strains are only found at subgingival crevice [126]. *Fusobacteria, Actinobacteria, Proteobacteria*, and *Cytophagales* are the four bacterial divisions mostly seen there but they hypothesized that more unknown taxonomic divisions are expected in this anatomical site.

Demmitt et al. performed another study where they used genome wide association studies (GWAS) for analyzing the oral microbiomes of 752 twin pairs [127]. They wanted to understand the genetic factors or identify the genes which regulate oral microbiome development. It was found that the host genes actually influence the oral microbiome to a great extent. The human genes were found on chromosome 7 and 12 which seemed to correlate with the oral microbiome while external factors like tobacco, marijuana and alcohol do not appear to have as strong an influence.

As one of the major diseases associated with the oral microbiome, periodontitis is a focal point of oral microbiome research. One study used shotgun metagenomic sequencing and phylogenetic diversity to understand periodontitis dynamics [128]. Additional keystone pathogen species like *Haemophilus haemolyticus*, *Prevotella melaninogenica*, and *Capnocytophaga ochracea* were identified which function in a similar way as *Porphyromonas gingivalis* (the main microbe causing periodontitis) in altering the oral microbiome. These three microbes are less abundant like *P. gingivalis* and promote inflammation by perturbing host-microbe interactions which try to inhibit the disease. Through the use of deep sequencing, microbes association with periodontitis can be suggested as well as a trend where health oral microbiomes are mainly composed of gram positive bacteria [43]. The shift to a diseased state and dysbiosis happens with accumulation of a broad range of gram negative bacteria, more complex metabolic interactions and increased availability of nutrients to the microbes due to damaged oral tissues. This is also reflected by a decrease in alpha diversity of the oral microbiome which is associated with periodontitis [128].

### 3.3. Modeling Methodologies

A small number of studies have used network analysis or genome scale modeling to study the oral microbiome. Network analysis was used to analyze the co-occurence mechanisms among oral microbes and gut microbes [114]. It was found that *Neisseria elongata* and *Granulicatella adiacens* were involved in causing metabolic syndrome (MetS). Currently, there are some conflicting results on the effects of external factors like diet, smoking, personal hygiene and medication on the oral microbiome [114,127] and it is expected that additional studies and analysis can help reconcile mechanisms lead to disease states. Another study performed metabolic reconstructions for 456 oral microbial strains using their genomic information [113]. Then, they applied network robustness analysis on the metabolic networks of these microbes to observe the biosynthetic capabilities of them. The function of 97 metabolites across the 456 microbial strains was determined to explore the cross-feeding and metabolic interdependencies among the oral microbes.

A genome scale metabolic model (GEM) of *P. gingivalis* has been reconstructed containing 679 reactions and 564 metabolites [112]. Analysis of this model indicated that *P. gingivalis* produces succinate, butyrate and propionate after amino acid catabolism. This is a strategy to harm host cells and feed its oral partners with essential nutrients. Furthermore, succinate produced by another oral pathogen is used to produce ATP. The GEM of *P. gingivalis* was also used to study synthesis pathways for lipopolysaccharides (LPS) which can potentially be targets for therapeutic applications [112]. *P. gingivalis* secretes gingipains which perform C5 convertase complement signalling with TLR2 protein leading to dysbiosis and inflammation in the normal oral microbiota [111]. The abundance of microbes present in a healthy microbiome gets disturbed due to the microbial dysbiosis caused by less abundant *P. gingivalis*. Hence, community metabolic modeling using GEMs involving this pathogen and its other oral partners will give us more answers to the biological dynamics of periodontitis which will help us in defining the therapeutics for this disease. 

## 4. Human Skin Microbiome

### 4.1. Background

With constant exposure with the outside world, the skin microbiome has the potential to be highly variable. Skin has different types and each type has its own microbiota composition and associated disease [129]. For sebaceous skin, *Corynebacterium* and *Propionibacterium* are found and associated with acne. Atopic dermatitis is found on moist skin and is associated with *Staphylococcus* and *Corynebacterium*. Psoriasis is a dry skin disease and species found there are *Corynebacterium*, *Propionibacterium*, *Staphylococcus* and *Streptococcus.* Given the diversity of skin types, diseases, and associated organisms one of the first needs is to identify and categorize the microbiota present. Bewick et al. tried answering these questions by performing trait based analysis on 971 skin microbial taxa to classify the organisms based upon various enzymatic or biological characteristics [130].

While generally showing high compositional variability, the skin microbiome shows higher variability between individuals with less variability in a single individual across different skin sites. Due to high interpersonal variations in these microbiomes, people have tried to study what are the skin microbes which actually differ between individuals [131]. 263 specimens were collected from 15 healthy adults (nine males, six females) from 18 defined anatomical skin sites. There were individual specific signature species in the skin microbiome which were found across all the skin sites in an individual but in low abundance. Sometimes the variation in the skin microbiome among different individuals is due to strain variations such as was found for different strains of *P. acnes* [131,132]. Genome sequencing and metagenomic studies of *P. acnes* showed that R2 and R6 strain of this bacterium are commensal ones whereas R4 and R5 strain showed strong link with acne [133]. They also identified the genetic determinants in *P. acnes* strains responsible for acne pathogenesis. Other than *P. acne*, *M. tuberculosis*, *M. leprae* and *S. aureus* are other bacteria which show this phenomenon [132]. *Staphylococcus epidermidis* is interesting as different strains of it are found at different body sites of an individual but the same strain of it has been found on different individuals. [131,132].

Atopic dermatitis causes dysbiosis in the skin microbiome which leads to an imbalance in the microbial populations consisting of *Propionibacterium, Corynebacterium* and *Staphylococcus epidermidis* which make allergens and pathogen associated molecular patterns (PAMPS) enter the skin easily [134]. To study atopic dermatitis, MiSH (Microbial index of skin health) was calculated for 25 skin bacterial genera from population samples from two Chinese cities and one American city [44]. For both these populations, it was observed that *S. aureus* is highly abundant in atopic dermatitis patients than the healthy patients indicating it as a global phenomenon [44,134].

In addition to site specific diseases, the skin microbiome plays a huge role in immunity. It prevents the invasion of the foreign pathogens, so, it is necessary to maintain the skin microbiome properly to live healthy [134,135]. Skin is not an ideal environment for microbes as they get low amount of nutrients, are always attacked by antimicrobial peptides and have an acidic, cool, salty and a desiccated atmosphere. Yet human and skin microbes have shown coevolution for their own benefit [135]. Primary immunodeficiencies (PIDs) are characterized by depleted memory T and B cells, changing eosinophil concentrations, high IgE levels and atopic dermatitis like skin disease [136]. In PID patients, *S. epidermidis*, *S. aureus* and *S. marcescens* were present in high abundance [136]. It was assumed that *S. epidermidis* may coordinate with *S. aureus* during PID in a mutualistic or commensalistic way enabling greater resistance by *S. aureus* to *S. aureus* antibiotics and other antimicrobial peptides. The skin of PID patients face selective pressures due to which different microbes try to take advantage of local changes such as *Serratia marcescens* being found on the skin of immunodeficient patients [136]. Another study showed that *S. epidermidis* can serve as a protective agent by inhibiting *S. aureus* biofilm formation [137]. The commensal form expresses a serine protease glutamyl endopeptidase (Esp) which degrades the proteins of *S. aureus* which they use for forming biofilm. This study also showed that *S. aureus* also switch to commensal form from pathogenic form when it interacts with *Corynebacterium striatum* [137].

### 4.2. High-Throughput Studies

The skin microbiome aids in protecting us from the invasion of the foreign pathogens, so, a great deal of biology is involved in the skin microbiome among the skin microbes and analyzing their metabolic interactions will give us several insights into the various skin diseases [138]. Use of amplicon sequencing and shotgun metagenomic sequencing has identified several fungal species (*Malassezia spp*., *Aspergillus spp*) and viral species (*Merkel cell polyomavirus* and *Molluscum contagiosum virus*) that are also present in skin microbiome but in very low abundance [137]. Generally, all experimental techniques (culturing, sequencing, etc.) are difficult due to the high variability that is possible in the skin microbiome including intrinsic factors like age, genetic make-up and host immune system and temporal changes in composition due to shifts in environmental exposure [139].

As with other microbiomes, 16S rRNA sequencing has been a primary method used to identify species associated with the skin microbiome. Due to the potential for variation of the skin microbiome on the same person, often multiple sampling methods can be used within the same study. For example three sampling methods: swab, scrape and punch biopsy were used to collect the samples from the inner elbow to study atopic dermatitis [140]. Various bacteria from *Proteobacteria, Actinobacteria, Bacteroidetes* and *Firmicutes* were found using the different sampling methods. Two indexes, Jabund and SONS assess the community similarity which showed that left elbow and right elbow had the same microbial community structure [140].

Interestingly, the skin microbiome appears to be temporally stable [141]. This time invariance was demonstrated using shotgun sequence data of skin microbes from 594 samples from 12 healthy individuals that were sampled across 17 skin sites at three time points. External and internal factors may play a crucial role in altering skin microbiome because without these perturbations, skin microbiome did not change at all [141]. There is a need to find how different factors perturb this stability and how the signature microbes in the skin microbiome maintain this stability. The human skin microbiome is very different from not only primates but from other mammals [142] with the human microbiome having dominant species of *S. epidermidis*, *Corynebacterium*, and *Propionibacterium acnes*. These species are not found in the skin microbiomes of other mammals. The reason for this distinction in the human skin microbiome is because of the general human practices which we do every day like clothing, living in a built environment and washing with soap etc. This has enabled the human skin microbiome to have a distinct species composition. This study was conducted using 16S rRNA sequencing targeting hypervariable regions (V3–V4) in the microbial RNA [142].

### 4.3. Modeling Methodologies

Network analysis was applied on skin microbiome to understand the topographical distribution of skin microbes at the three skin sites: moist, sebaceous and dry areas [115]. Each of these sites has their own microenvironment with their own set of species and metabolites. Moist skin microbes outnumber sebaceous skin microbes and moist and sebaceous microbe members frequently group together depicting high number of metabolic interactions happening between these two communities [115].

Bottom-up ecology is a biological phenomenon which says that for a microbiome, the metabolic potential of it which is depicted by the ability to use the metabolic resources effectively can be known from genes, metabolites and interactions among them in the species [116]. This study used genome scale modeling (GEM) to investigate this concept in skin microbiome. They found the top 10 metabolites/compounds which are present in higher amount in the skin microbiome. These were the ones which are regularly found in cosmetics and hygiene products like myristic acid, citrate, *N*-acetyl glucosamine etc. [116]. These results show that external factors somehow do influence the skin microbiome to a great extent. 

## 5. Human Vaginal Microbiome

### 5.1. Background

Vaginal microbiome research is the most new one if compared with microbiome research in the other three anatomical sites; gut, oral and skin. Currently, more than 100 to 200 bacterial species have been found associated with the vaginal microbiome [143]. *Lactobacillus* (*L.*) is the dominant genera and is typically associated with a healthy vaginal microbiome. They are responsible for the low pH we usually see in this microbiome, i.e., 3.8 to 4.5 [143] which helps prevent proliferation of nonindigenous organisms and helps in defense against the foreign pathogens that can cause dysbiosis and vaginal infections [144]. Interestingly, even though *Lactobacilli* are common in the vaginal microbiome, the ethnicity of an individual appears to influence the specific species with *L. iners* found in White/Caucasian people, *L. crispatus* in Asian women and *L. jenseni* found in Black and Hispanic women [143]. However, a significant number of healthy women also have vaginal microbiome which is not dominated by *Lactobacillus* species with 40% of Black and Hispanic populations having non-dominated lactobacilli vagitypes [144,145]. Even in non-*Lactobacillus* dominated microbiomes, other bacteria perform homolactic and heterolactic acid fermentations indicating the importance of lactic acid production.

The species distribution in the vaginal microbiome can be consistent across individuals as similar vagitypes have been identified across individuals, which is not the case for other microbiomes as we mentioned above. Community state types (CSTs) which have been classified according to the dominant species found within a vagitype. CST1, CST2, CST3, and CST5 are vagitypes dominated by *Lactobacillus* species. CST1 comprise *L. crispatus* as dominant species, CST2 is dominated by *L. gasseri*, CST3 consists of *L. iners* as dominant species and CST5 has *L. jenseni* [146,147]. CST4 is dominated by a diverse group consisting of anaerobic bacteria: *Prevotella, Dialister, Atopobium, Gardnerella* etc. with low proportions or no detectable *Lactobacillus* species [144]. Vaginal bacterial communities dominated by *Lactobacillus spp.* i.e., CST1, CST2, CST3 and CST5 were found in 80.2% and 89.7% of Asian and white women, respectively and in 59.6% and 61.9% of black and Hispanic women, respectively. Communities with low proportions or no detectable *Lactobacillus* species community state type, i.e., CST4 were elevated in Hispanic (38.1%) and black (40.4%) women compared with Asian (19.8%) and white (10.3%) women [144,146]. People have also defined some other CST types i.e., CST6 which is dominated by *G. vaginalis*, CST7 comprise BVAB1 (Bacterial Vaginosis associated bacteria-1); CST9 consists of *G. vaginalis* and *L. iners* as dominant species [147].

A shift in the composition of the microbiome leads to dysbiosis and potentially bacterial vaginosis (BV) that is characterized by a shift away from *Lactobacilli* and a change in the properties of the vaginal fluid [143]. Clinically, bacterial vaginosis is characterized by a Nugent score between 7 to 10 and can be a cause of preterm birth [42,148]. BVAB1 and *G. vaginalis* are currently considered the main causative agents for bacterial vaginosis and their presence often coincides with an increase in the local pH [145]. There are a set of intrinsic and extrinsic factors which are responsible for causing this dysbiosis. Internal factors are genes, innate and adaptive immune mechanisms and external factors are antibiotics, hormonal contraceptives, birth control methods, sexual activity, vaginal lubricants and douching which can cause disturbance to this microbiome [145]. CST4 is considered as the vagitype which has the highest ability to cause bacterial vaginosis [146]. *L. iners* has been associated with dysbiosis as it has been observed that *L. iners* number increase in a dysbiotic vaginal microbiome [143]. CST3 is the vagitype dominated by *L. iners*. It has been observed that CST6 and CST9 which are BV associated microbiomes, transitioned to CST3 after BV medication was taken [147]. Hence, medication decreased the risk, but the treatment was not fully effective. It is because *L. iners* use the high pH of a dysbiotic microbiome to perform cooperative interactions with BVAB1. CST1, CST2 and CST5 did not show any kind of transition even after BV medication which shows that if the microbiome is healthy it has a certain degree of robustness to change [147].

Ethnicity of a person has been associated with a particular type of vaginal microbiome. The microbiomes in African-American women are more diverse and with a higher number of BV-associated bacterial species such as *G. vaginalis*, *L. iners*, BVAB1, *Mycoplasma hominis*, *Aerococcus* and other anaerobes like *Anaerococcus* [148]. Caucasian women were more likely to be colonized with *L. crispatus*, *L. jensenii*, *L. gasseri* and *Staphylococcus*. In a comparison of African-American and European women, it was found that African-American women are 2.9 times more likely to be diagnosed with BV and European women had 25.8% less BV-associated bacteria [148]. Pregnant women who were African-American showed higher risk of preterm birth than European pregnant women [148].

### 5.2. High-Throughput Studies

As with most of the human microbiome projects, 16S rRNA sequencing has been used for initial analysis and identification of microorganisms, such as the identification of the dominant *Lactobacillus* species through measurement of samples from women from different countries [149]. Studies have also been conducted to determine if the vaginal microbiome has variation in composition within the same person. For example, 10 Finnish women had bacterial DNA from 6 different vaginal locations using 4 different sampling strategies tested and generally, the microbial composition was shown to be consistent regardless of sampling site or strategy [150]. Based upon 16S rRNA sequencing, a vaginal microbiome database (Vaginal 16S rDNA Reference Database) has been created containing information of bacterial species in the vagina [151]. This database contains information from 1017 mid-vagina samples and approximately 30 million 16S rDNA reads. Species classification was based on hypervariable regions from V1 to V3 in the 16S rDNA using the STIRRUPS algorithm [151].

More recent studies have sought to leverage high-throughput sequencing to gain additional insight into the microorganisms in the vaginal microbiome [42,152]. For example, vaginal microbiome samples from 310 healthy Canadian women were analyzed using massively parallel sequencing of the universal target (UT) of the cpn60 gene [152]. Six community state types (CSTs) were found where groups 1, 3 and 5 were as previously discussed, composed primarily of *L. crispatus*, *L. iners* and *L. jensenii*, respectively but group 4 (CST4) was explored more. CST4 was divided into two subgroups on the basis of *Gardnerella* subgroups. Individuals with dominant *Gardnerella* subgroups A and C separated into CST IVC and IVD, respectively, while those with dominant *Gardnerella* subgroup B clustered within the heterogeneous CST IVA category [152]. In addition to gaining better insight into *Gardnerella,* another study has used sequencing of the V4 region in the 16S rRNA genes to study the role of *Prevotella* [153]. It was observed that *Prevotella* may act as the intermediate link between host genetics and vaginal microbiome dynamics with *Prevotella* affecting the host metabolism or regulating the functions of host genes. Furthermore, *Prevotella* is found in 72.2% of people’s vaginal microbiome which is even higher than *L. crispatus* (36.9%) and *L. iners* (41.2%) [153].

Currently, the largest vaginal microbiome study analyzed 597 pregnancies and ~12000 samples [42]. The primary outcome was the identification of the signature vaginal microbes associated with preterm birth (PTB): BVAB1, *Prevotella* cluster 2 (*Prevotella timonensis* and *Prevotella buccalis*), *Sneathia amnii*, *Megasphaera* type1 and TM7-H1. PTB signatures were present more in the cohort of African American women and identification of these species early pregnancy may help predict the likelihood of PTB [42]. This large scale data set also analyzed the vaginal microbiome during pregnancy of Caucasian and African-American women [118]. Overall, alpha diversity in vaginal microbiome of non-pregnant women was higher than the pregnant women. In African-American women, the vaginal microbiome shifted towards a more *Lactobacillus* dominated profile from a highly diverse starting microbiome over the course of pregnancy [118]. Metabolic simplification happens during the transition where diversity decreases and most of the pathway reduction happens in first trimester in African pregnant women [118].

### 5.3. Modeling Methodologies

Currently, very few modeling studies exist for the vaginal microbiome. Constraint-based methods were used to reconstruct genome scale metabolic models (GEMs) for the several of the organisms identified in the vaginal microbiome: TM7-H1, BVAB1 and *L. crispatus* [42,118]. They used functional annotation information of these organisms with Enzyme Commission numbers to describe function and KEGG IDs for nomenclature. Another study has employed a community metabolic modeling approach where species-specific genes are considered based upon metagenomic data in the healthy and diseased state [117]. These genes are then paired with metabolic reactions to find the community metabolic potential (CMP) score for every metabolite in each sample [117]. CMP score variation shows the variation in the metabolite abundances which helped them to identify the crucial metabolites. Then, species, genes and reactions are identified which were responsible in giving that CMP score for a metabolite. Bacterial vaginosis associated species (*Eggerthella sp. 1*, *Megasphaera type 1*, and *Mageeibacillus indolicus*) have a substantial impact on the vaginal microbiome and were designated as “drivers” of species-metabolite dynamics which caused metabolic variation in the vaginal microbiome [117].

## 6. Perspectives and Future Work

The human microbiome is an ecological setting where different microbial species are interacting with each other through metabolite and protein exchange and is dynamic and complex in nature [3]. The diversity of microorganisms currently identified is high (Figure 2) and the relationships between organisms can be either cooperative or competitive where microbes exchange the essential elements for their own benefit or the dominant microbe may try to exploit the maximum resources using its own metabolic network and secrete inhibitory agents against other microbes [154]. These interactions form the basis of the working of the entire human microbiome. From these points, it is clear that human microbiome is not solely characterized or dependent on the working of a single microbe or single species [34,155], but rather it must be studied as an interacting community. One approach that can facilitate studying the human microbiome as an interacting, dynamic community is community genome scale modeling [86]. However, challenges exist to utilizing genome scale modeling (GEM) including the low number of models for human microbiome species and the need for further improvements in computational frameworks for large-scale microbial communities (including appropriate objective functions and compartmentalization issues) [28,35,154]. Ongoing research is working on identifying and annotating novel human microbiome species which is needed as a basis for metabolic modeling. Furthermore, once models are generated, details on material (protein, metabolite) exchange and appropriate community objective functions need to be considered [6,21,36]. Advances in this manner will aid in understanding the integrated function of the microbiome (interactions and temporal dynamics) to give a better perspective of the functional microbiome [26,28,35].

The gut microbiome is the human microbiome with the largest number of species with greatest diversity involved in many diseases including obesity and diabetes [72,73] [98]. Even though much effort has gone into studying the gut microbiome, half of the total number of microbes in the gut microbiome have not been annotated even at the genus level [68,78,82]. Additional knowledge of species and molecular functions for obese and diabetic microbiomes will enable us to understand the host and microbial metabolism aspects in a more detailed manner [49]. Disease, diet, geography, lifestyle and even occupation are responsible for giving a unique gut microbiome for every individual as these factors hugely alter the microbiome [48,60,66]. These alterations happening in this large-scale system can be explored only by investigating the metabolic interactions among the microbes, microbe and diet/host and for that, community-based genome scale metabolic modeling may help once annotated species are identified [27,36]. Furthermore, the gut microbiome is involved in many biological functions like carbohydrate metabolism, colonization resistance, and ROS production during cancer [39,53,64]. These biological processes individually are comprehensive in nature and both experimental and computational investigations to the gut microbiome during each of these processes will give us many novel findings and wide range of medical solutions.

Oral microbes live and form stable metabolic interactions in the biofilms [119,120]. Similar to the gut microbiome, the oral microbiome is large (700+ different species) and has a large number of unannotated species (between 50% [119] and 76% [123] unannotated). One reason for this low level of species characterization in the oral microbiome is because oral microbes often form biofilms making isolation and sequence analysis difficult. Thus, mechanisms of biofilm formation, maintenance and dynamics need to be considered in studying the oral microbiome and oral diseases [125]. In addition, novel biological mechanisms such as peptidic small molecules (PSMs) have been identified and are responsible for giving structure to the oral microbiome [124]. Microbe–microbe interactions have been explored less for this oral microbiome using modeling methods. *Corynebacterium*’s presence at biofilm matrix for structuring the microbiome or *P. gingivalis*’s strategies during periodontitis occurs only through stable metabolic interactions with their oral partners [119,121]. Performing GEM for just *P. gingivalis* therefore will not give us the entire picture of the etiology of periodontitis as Benn et al. suggested that it is never a single oral pathogen causing an oral disease [112,119]. Therefore, focus of oral microbiome research should shift towards species identification and microbial interactions using both experimental and computational methods. The oral microbiome, due to its topological position, is heavily affected by food, smoke, alcohol, drugs and medication [114]. Genome scale models (GEM) if performed to see these effects on oral microbiome will give us important therapeutic inferences. For example, we can use GEM to investigate microbial metabolites and nicotine metabolites interactions to assess the smoking effects. This type of study might give us the oral species utilizing the tobacco metabolites which can be further used as drug targets to decrease the harmful smoking effects.

Skin microbes produce antimicrobial peptides against the foreign pathogens making the skin microbiome highly involved in the immune mechanisms of the host [134,135]. Therefore, performing detailed investigations on the skin microbiome can give us the idea on more kinds of immune strategies taken by skin microbes. The dermatological disorders like acne, psoriasis, atopic dermatitis or primary immunodeficient diseases (PIDs) are all caused by a particular bacterial trigger of host skin species [129]. These human health impacts occur due to the interesting property shown by skin microbes; switching mechanism. Sebaceous skin harbors *Propionobacterium acnes* which switch from a host protective commensal strain to a host damaging pathogenic strain causing acne [133]. Similarly, *Staphylococcus epidermis* and *Staphylococcus aureus* also show these switching mechanisms to pathogenic strains making them the causal agents for atopic dermatitis and PIDs [137]. Research has identified this switching behavior of skin microbes but mechanisms associated genes, proteins, or pathways which the microbes use to perform the switch have not been found yet. *Serratia marcescens* is another microbe found in the skin microbiome during PIDs [136] and a genome scale model for this microbe has been reconstructed [156] making it a candidate organism for studying microbial switching mechanisms. The skin microbiome has been proved to be temporally highly stable and shows high interpersonal variation [131,141]. Therefore, more time course studies to find out when exactly the skin microbiome gets stable and unique in an individual should be performed. External factors significantly affect the skin microbiome as it has been shown that the top 10 metabolites which affect the microbiome came from cosmetics [116]. Skin microbiome’s research focus should move towards exploring the strange mechanisms associated with this microbiome like high stability, high interpersonal variation, switching mechanisms of the microbes and external effects. Albert Kligman and Donald Pillsbury, two renowned dermatologists at the University of Pennsylvania in 1954 stated “a great deal more remains to be learned about the forces which control the bacterial ecology of the surface of the skin” [138]. This remains as true today as when this was originally stated in 1954.

The vaginal microbiome behaves differently than other microbiomes as consensus is starting to identify core species found in the majority of healthy women, i.e., *Lactobacillus* [143]. However, a diverse microbiome where *Lactobacillus* does not dominate has also been found in many healthy women so it is still difficult to predict the effects of the microbiome based upon the dominant species. For example, CST3 is a vagitype dominated by *Lactobacillus iners,* but CST3 is prone to bacterial vaginosis because *L. iners* interacts with the main causative agent of this disease; BVAB1 [143,147]. Hence, having *Lactobacillus* dominated microbiome does not mean that person is not at risk, but a high diversity vaginal microbiome almost always has a greater chance to transition to a diseased microbiome [143,144,145]. Specific to the vaginal microbiome, evidence is arising demonstrating that ethnicity may influence the vaginal microbiome composition. For example, the highly diverse vagitype CST4 is found in 40% of black and Hispanic population making them more prone to bacterial vaginosis and preterm birth [42,144,146]. The reason behind this disparity is currently not known. We have a better understanding of what species are present in the vaginal microbiome during pregnancy, preterm birth and bacterial vaginosis [42,118,148], but we need to know what functions individual organisms are performing during these healthy and disease states, what type of metabolite exchanges are happening between them, what kind of relationship, i.e., commensal or cooperative or competitive is shown by them. The answers to these questions can be obtained by performing more GEM studies on multiple vaginal microbes in their CSTs [42,118].

## 7. Conclusions

Human microbiome is a vast biological system where a great deal of knowledge is still unknown including novel species and genes. The four main anatomical sites of the human microbiome (oral, skin, gut and vaginal) harbor different sets of species and the species composition and the metabolic interactions among them changes in response to many different variables, both intrinsic and extrinsic. Furthermore, the species composition is also different for different individuals for all the microbiomes. The complexity and variability of the human microbiome make it a potentially daunting area of study, but the significance of the impacts of the human microbiome on human health warrant focused research attention. While we are still at the phase of identifying and understanding the constituent components of the human microbiome, it is necessary to push development of experimental and computational methods that can help understand the large, interconnected, and dynamic system of the human microbiome. Research studies in this direction will definitely help us to make bigger impact on public health by creating effective medicinal strategies. It is expected that computational analyses will be necessary to study the human microbiome to gain insight to the various metabolic mechanisms of the microbe, its interaction partners, and if scaled up, the microbial niche at a particular anatomical site and someday, the whole human microbiome.

## Figures and Tables

**Figure 1 microorganisms-08-00197-f001:**
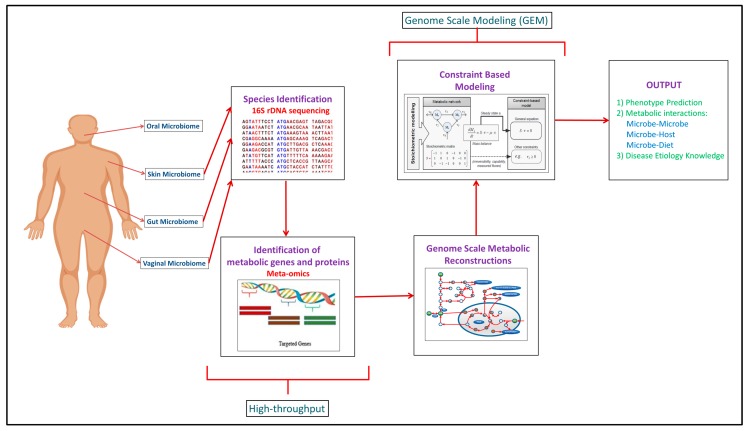
Complete human microbiome research pipeline established over the years explaining the integration of experimental and computational methodologies to get the mechanistic understanding of the human microbiome.

**Figure 2 microorganisms-08-00197-f002:**
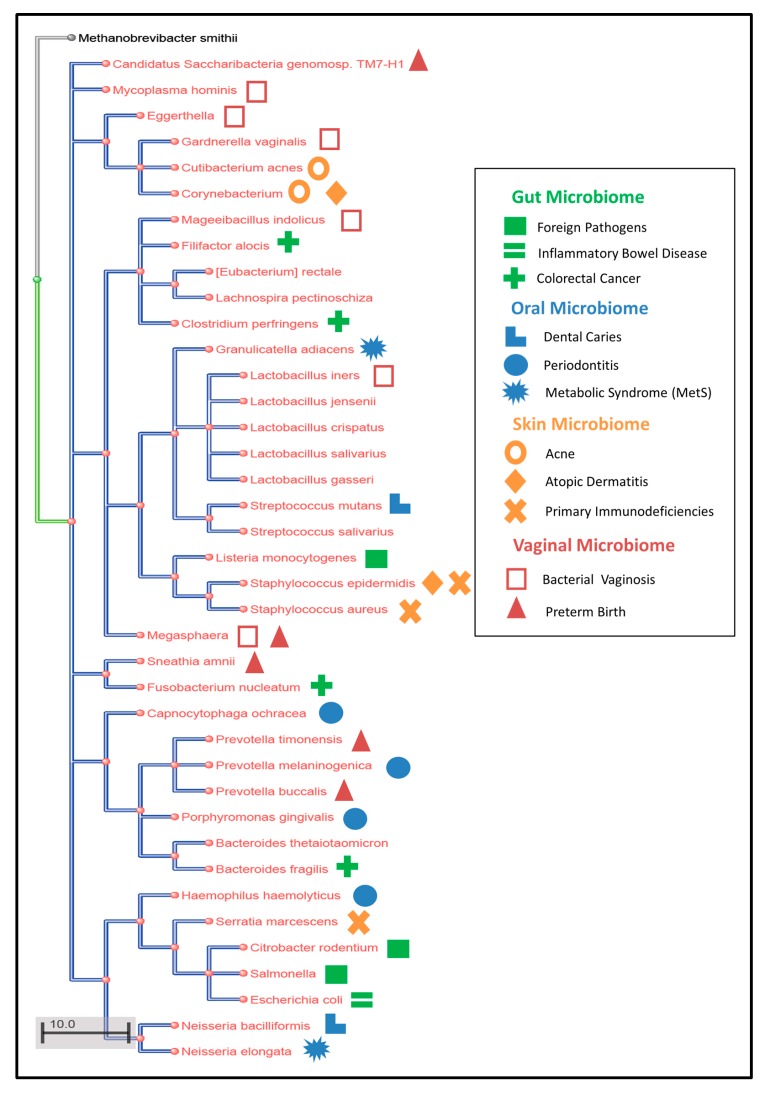
Phylogenetic tree for the all the human microbes present at the 4 antomical sites (gut, oral, skin, vagina). The disease caused by the resident human microbes is shown by a particular shape.

**Table 1 microorganisms-08-00197-t001:** Summative table of all the modeling studies performed on Gut, Oral, Skin and Vaginal microbiome during disease and non-disease scenarios.

Non-Disease	Disease	Experimental Data	Modeling Approach	Reference
	**GUT MICROBIOME**			
Microbe–host–diet interplay		Meta-omics	Genome Scale Modeling	[99]
Virtual Metabolic Human Database		Experimental data from published articles (16S rRNA sequencing, Meta-omics etc.)	Genome Scale Modeling	[103]
Gut microbiome dynamics in infants and small children		Metagenomics	Genome Scale Modeling	[102]
Host–food–microbiome interplay		Omics data of various types	Genome Scale Modeling	[95]
Spatial distribution of metabolites at gut microbiome		Microbial culturing	Genome Scale Modeling	[105]
Microbe–microbe and host–microbe interactions		Metabolomics, Transcriptomics, Proteomics	Genome Scale Modeling	[96]
Gut microbiota dynamics		16S rRNA sequencing, Metagenomics	Genome Scale Modeling	[91]
Gut host-microbe metabolism		Metagenomics	Genome Scale Modeling	[101]
	Obesity	Metagenomics	Network Based Approach	[24]
	Inflammatory bowel disease	Metagenomics	Network Based Approach	[24]
Diet’s influence on gut microbiome		Metabolomics	Community Metabolic Modeling	[90]
Gut microbiome dynamics		Omics and Meta-omics	Genome Scale Modeling	[98]
Gut microbiome dynamics		Single-cell genomics, Metagenomics, Metatranscriptomics	Genome Scale Modeling	[22]
Gut microbiome metabolism		Genomics, Proteomics, Metabolomics	Genome Scale Modeling	[97]
Host-microbe interactions		Omics data of various types	Genome Scale Modeling	[100]
Gut microbiota dynamics		Whole-genome sequencing, Transcriptomics, Proteomics	Genome Scale Modeling	[109]
Gut microbiome dynamics		Culturomics, Next-generation sequencing (NGS), Phenomics, Transcriptomics, Metabolomics, Proteomics	Genome Scale Modeling	[84]
Diet-microbe, microbe–microbe and host-microbe interactions		Meta-omics, Metabolomics	Genome Scale Modeling	[83]
Gut metabolome investigation		Metabolomics	Genome Scale Modeling	[110]
*E. coli* metabolism		Deep sequencing/NGS	Genome Scale Modeling	[85]
	Malnutrition	Metabolic profiling/Metabolomics	Genome Scale Modeling	[87]
Spatial and temporal dynamics		Metagenomics	Agent Based Modeling	[23]
	Autism	Metabolomics	Genome Scale Modeling	[106]
	Colon cancer	Omics data of various types	Genome Scale Modeling	[107]
	Colorectal cancer	Metabolomics	Genome Scale Modeling	[108]
	**ORAL MICROBIOME**			
	Periodontitis	Transcriptomics	Genome Scale Modeling	[111]
	Periodontitis	Genomics	Genome Scale Modeling	[112]
Oral microbiota dynamics		Metabolomics	Network Based Approach	[113]
	Metabolic Syndrome (MetS)	16S rRNA sequencing	Network Based Approach	[114]
	**SKIN MICROBIOME**			
Topographical dynamics at skin microbiome		16S rRNA sequencing	Network Based Approach	[115]
Skin microbiome dynamics		Metagenomics	Genome Scale Modeling	[116]
	**VAGINAL MICROBIOME**			
Community dynamics at vaginal microbiome	Bacterial vaginosis (BV)	Metabolomics	Community Genome Scale Metabolic Modeling	[117]
	Preterm birth	16S rRNA sequencing, Metagenomics, Metatranscriptmics	Genome Scale Modeling	[42]
Vaginal microbiome dynamics during pregnancy		16S rRNA sequencing, Metagenomics, Metatranscriptmics	Genome Scale Modeling	[118]

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
