# Peer review of "Computational Modeling of the Human Microbiome"

_microorganisms, 2020, doi:10.3390/microorganisms8020197_

Round 1

Reviewer 1 Report

The manuscript summarized applications of various sequencing and modeling approaches in the research of human microbiome. Entitled with "computational modeling", however, description of the modeling part takes only a small portion of this review.

As below are some of my other concerns: 

1 Lack of systematic introduction of sequencing approaches like amplicon and metagenomics.

2 Species identification is a difficult and critical problem in metagenomics, reads assignment algorithms (binning) should be introduced in the manuscript.

3 In line 213, author mentioned the limitation of read length, I wonder is there any application of 3rd generation sequencing in gut microbiome.

Modeling methodologies should be the emphasis which is, however, full of description of applications. Basic principles are required for these methods.

5 Category of modeling approach in Table 1 is obscure. For example, what's the difference between Genome Scale Modeling and Genome Scale Metabolic Reconstruction? And GENREs is a Constraint Based Modeling approach in principle.

6 I wonder if it is necessary to divide the manuscript into four parts according to anatomical sites at the beginning since the sequencing and modeling methods are largely the same with the exception of various microbial species and their abundance.  

Author Response

1) Lack of systematic introduction of sequencing approaches like amplicon and metagenomics.
Answer: We thank the Reviewer for this suggestion and have revised the manuscript to include background information on relevant sequencing methods in the Introduction including discussion on 16S rRNA sequencing (page 2 lines 54-64), metagenomics or meta-omics approaches (page 2 lines 69-78) and the role of meta-omics approach in regard to genome scale modeling (page 2 lines 82 to 96)

2) Species identification is a difficult and critical problem in metagenomics, reads assignment algorithms (binning) should be introduced in the manuscript.
Answer: As the Reviewer rightly indicates, species identification is an important issue with metagenomic data. We have expanded the discussion of this topic to highlight this issue (page 6/7 lines 279-286).

3) In line 213, author mentioned the limitation of read length, I wonder is there any application of 3rd generation sequencing in gut microbiome.
Answer: Yes, third generation sequencing has recently been applied to study the gut microbiome and partially addresses some of the problems associated with limited read lengths. This has been added to the revised manuscript (page 6 lines 264-268).

4) Modeling methodologies should be the emphasis which is, however, full of description of applications. Basic principles are required for these methods.
Answer: We agree with the Reviewer’s suggestion and have added a new paragraph to the Introduction section to present the basic principles of Genome scale metabolic modeling (GEM) (page 3 lines 104 to 127).

5) Category of modeling approach in Table 1 is obscure. For example, what's the difference between Genome Scale Modeling and Genome Scale Metabolic Reconstruction? And GENREs is a Constraint Based Modeling approach in principle.
Answer: We apologize for the confusion and there is no difference between Genome Scale Modeling or Genome Scale Metabolic Reconstruction or Constraint Based Modeling. Different authors and publications use slightly different terminology (and acronyms) to refer to the same modeling approach. Table1 now uses a single term of genome scale modeling (GEM) to denote the modeling approach. This confusion has been resolved in the paper also where we are using only one name for the modeling approach i.e. genome scale modeling (GEM) and in few points, we used genome scale metabolic models (GEM) but the abbreviation is same i.e. GEM denoting same meaning.

6) I wonder if it is necessary to divide the manuscript into four parts according to anatomical sites at the beginning since the sequencing and modeling methods are largely the same with the exception of various microbial species and their abundance.

Answer: We considered the Reviewer’s suggestion and currently would prefer to keep the manuscript organized where general principles or concepts that apply to all human microbiomes are presented in a common Introduction and site-specific content is contained in the individual sections. We believe this will help streamline the manuscript by reducing redundancy.

Reviewer 2 Report

In the review “Computational modeling of the human microbiome” the authors summarize the microbiome data from four anatomical sites (gut, mouth-oral, skin, vagina), providing information on site-specific background, experimental data and computational modeling.

Reviewer’s points

As the authors recall in their introduction the researchers who face the study of the microbiome must deal with numerous difficulties, high inter- and intra-personal variability caused by several factors (i.e. environmental conditions, host-genetics), microbe-microbe and microbe-host interaction, identification of low abundant species and, above all, the most suitable bioinformatic tools to be used to obtain more information from their experimental data. Therefore, an in-depth study of the most current bioinformatic approaches to obtain information useful for a greater knowledge of the microbiome role in a given human district and what the influence on the host’s metabolism could be very useful. However, in order to reach their goal towards the readers (not all expert in microbiome), the authors should deepen the “Modeling methodologies” paragraphs 2.3, 3.3, 4.3 and 5.3 by adding  to each bioinformatic modeling which experimental data (16S rRNA, metagenome, metatrascriptome, metabolome, etc) are needed (to add such information in a table is also suggested).

Minor points

-pag.2, line 74 correct as indicated “…analysis (FBA)…”

-pag.4, line 137 “…diet had a minimal effect in altering….”, add a reference.

-pag.7, line 284 “…using the Recon 2….”, add a reference. Further, following the release of Recon 2 several updates were published which captured human metabolism more accurately, please cite some of the most recent references.

-pag.8, line 351 correct as indicated “….during disease and non-disease scenarios are…”.

Author Response

 1) As the authors recall in their introduction the researchers who face the study of the microbiome must deal with numerous difficulties, high inter- and intra-personal variability caused by several factors (i.e. environmental conditions, host-genetics), microbe-microbe and microbe-host interaction, identification of
low abundant species and, above all, the most suitable bioinformatic tools to be used to obtain more information from their experimental data. Therefore, an in-depth study of the most current bioinformatic approaches to obtain information useful for a greater knowledge of the microbiome role in a givenhuman district and what the influence on the host’s metabolism could be very useful. However, in order to reach their goal towards the readers (not all expert in microbiome), the authors should deepen the “Modeling methodologies” paragraphs 2.3, 3.3, 4.3 and 5.3 by adding to each bioinformatic modeling which experimental data (16S rRNA, metagenome, metatrascriptome, metabolome, etc) are needed (to add such information in a table is also suggested).
Answer: We thank the Reviewer for their suggestions. We agree with the Reviewer’s suggestion and have added a new paragraph to the Introduction section to present the basic principles of Genome scale metabolic modeling (GEM) (page 3 lines 104 to 127).

The Reviewer’s second suggestion to include linkage to experimental data associated with modeling studies has been integrated into Table 1. The experimental data used in the modeling studies mentioned in Table 1 are added as a separate column. Corresponding text has also been added.

2) Minor points
-pag.2, line 74 correct as indicated “…analysis (FBA)…”
Answer: This line got removed.
-pag.4, line 137 “…diet had a minimal effect in altering….”, add a reference.
Answer: Reference added (line 184 to 187)
-pag.7, line 284 “…using the Recon 2….”, add a reference. Further, following the release of Recon 2 several updates were published which captured human metabolism more accurately, please cite some of
the most recent references.
Answer: Reference added for Recon 2, Recon 2.2 and ReconMaps (line 347 to 348). Also, reference for Recon 3d and ReconMaps added at a new place (line 375 to 380).
-pag.8, line 351 correct as indicated “….during disease and non-disease scenarios are…”.
Answer: Corrected (line 413 to 415).

Round 2

Reviewer 1 Report

I'm afraid the 4th concern is not well addressed after the author's revision. I still don't see how the modeling methodologies work for a better understanding of human microbiome. I think the readers need to know at least the basic principles of the tools, not just the enumerated applications. For example, the author mentioned a GutLogo model which is "developed to study the spatial and temporal dynamics". Is it built based on dFBA? What's an agent based model, and how it performs a perturbation test? Network analysis was also mention several times, but I didn't see any explanation or description of that.

Author Response

We thank the Reviewer for pointing this out and apologize for the omission in the previous drafts.  In the revised manuscript, we have added brief descriptions of network analysis and agent-based modeling.  We have also added a short sentence describing the general utility of using computational approaches to study the human microbiome.  The new text is found on page 3, lines 102-112 highlighted in yellow in the revised manuscript.

Reviewer 2 Report

The revision greatly improved the paper that is now suitable for publication.

Author Response

We thank the Reviewer for their constructive comments during the review process.

Round 3

Reviewer 1 Report

I think my concern is generally addressed.